# Long-lived metabolic enzymes in the crystalline lens identified by pulse-labeling of mice and mass spectrometry

Pan Liu[1], Seby Louis Edassery[2], Laith Ali[2], Benjamin R Thomson[1], Jeffrey N Savas[2]*, Jing Jin[1]*

[1]Feinberg Cardiovascular and Renal Research Institute, Feinberg School of Medicine, Northwestern University, Chicago, United States; [2]Department of Neurology, Feinberg School of Medicine, Northwestern University, Chicago, United States

**Abstract** The lenticular fiber cells are comprised of extremely long-lived proteins while still maintaining an active biochemical state. Dysregulation of these activities has been implicated in diseases such as age-related cataracts. However, the lenticular protein dynamics underlying health and disease is unclear. We sought to measure the global protein turnover rates in the eye using nitrogen-15 labeling of mice and mass spectrometry. We measured the $^{14}N/^{15}N$-peptide ratios of 248 lens proteins, including Crystallin, Aquaporin, Collagen and enzymes that catalyze glycolysis and oxidation/reduction reactions. Direct comparison of lens cortex versus nucleus revealed little or no $^{15}N$-protein contents in most nuclear proteins, while there were a broad range of $^{14}N/^{15}N$ ratios in cortex proteins. Unexpectedly, like Crystallins, many enzymes with relatively high abundance in nucleus were also exceedingly long-lived. The slow replacement of these enzymes in spite of young age of mice suggests their potential roles in age-related metabolic changes in the lens.

**\*For correspondence:**
jeffrey.savas@northwestern.edu (JNS);
jing.jin@northwestern.edu (JJ)

**Competing interests:** The authors declare that no competing interests exist.

## Introduction

The lens is a transparent body with an essential role in visual acuity. It consists of an outer capsule of type IV collagen-laminin membrane, the cortex of lens epithelium, and denuclearized and organelle-free fiber cells at the core. A single layer of germinal cells beneath the anterior capsule gives rise to transitional cells which differentiate into elongating fiber cells and finally the mature lenticular fiber cells forming the nuclear mass of the lens. In this process, the human lens continues to grow slowly in its weight and size throughout life (*Vavvas et al., 2002*; *Guirou et al., 2013*; *Augusteyn, 2007*; *Bassnett, 2002*). Frequently associated with aging, lens disease, such as cataracts, account for approximately half of the global blindness (the World Health Organization data). However, the underlying molecular mechanisms for most cataracts remain poorly understood (*Pescosolido et al., 2016*). Evidence suggest that accumulation of oxidized proteins and lipids predisposes the lens to nuclear cataract development (*Reddy, 1971*; *Williams, 2006*). Despite the slow turnover of the lens tissue, it remains a site of biochemical activity (*Reddy and Giblin, 1984*; *Hejtmancik et al., 2015*), in which the production of reducing metabolites and perhaps local enzymatic reactions within the fiber cells are important in combating oxidative stress.

Previously, radiocarbon ($^{14}C$) dating studies demonstrated that with respect of the total protein and lipid, there was little turnover at the nuclear core of lens (*Lynnerup et al., 2008*; *Hughes et al., 2015*; *Nielsen et al., 2016*). However, a recent single-fiber-cell transcriptome study of postnatal day 2 mice detected mRNA encoding proteins known to exist in anucleate lens fibers (*Gangalum et al., 2018*), implicating active protein synthesis. In order to gain insight into the protein dynamics of

lenticular fiber cells, including their structural proteins, chaperones and enzymes, we performed whole organism pulse-labeling of mice with heavy nitrogen-15 ($^{15}$N) supplied in the diet to assist distinguish newly synthesized proteins from their counterparts existing before labeling.

## Results

### The lens had extremely slow turnover of its proteome as compared to non-lens tissues in the eye

Between 3 and 15 weeks of age, mice were fed with exclusively $^{15}$N chow (*Savas et al., 2012*; *Liu et al., 2018*), following which the eye tissues were harvested. The entire lens was processed and then analyzed by liquid chromatography-tandem mass spectrometry (LC-MS/MS) (*Figure 1A–C* and Methods). In total, 535 proteins were identified at 1% FDR at the protein level, of which 248 proteins showed the presence of both fully $^{14}$N and $^{15}$N spectra for calculating their $^{14}$N/$^{15}$N ratios (MS1 ratios of the old vs. the new protein. See Methods, *Figure 1D* and *Supplementary file 1*). These ratios reflected the proportions of individual proteins being replaced with new $^{15}$N proteins in addition to newly formed fiber layers being added to the growing lens. More than 50% of individual proteins in the lens were only detected in their $^{14}$N forms, as compared to 1.8% and 2.8% in the vitreoretinal and sclera/choroid tissues respectively (*Figure 2A* and *Supplementary file 1*). These non-lens tissues had a majority of individual proteins completely replaced by new $^{15}$N proteins, in contrast to the slow turnover of 1.7% of individual lens protein. While the determination of lack of $^{15}$N peptides is subjected to MS detection sensitivity, we listed only the most abundant proteins with the total absence of $^{15}$N labels (*Figure 2B*). These extremely long-lived proteins included structural proteins such as PE-binding protein 1 (PEBP-1), β-Catenin, Moesin, and enzymes that are involved in oxidoreduction such as Peroxiredoxin-2 (Prdx2), Farnesyl pyrophosphate synthase (FPS) and Aldehyde dehydrogenase (Aldh7A), and in glycolysis such as ATP-dependent 6-phosphofructokinase (ATP-PFK). For those proteins with both $^{14}$N and $^{15}$N peptides detected, lens proteins exhibited the most long-lived proteins with greater $^{14}$N proportions (*Figure 2C*).

### The longevity of proteins associated with fiber cell differentiation

Meanwhile, the histone variant H3.3 was among the fastest turned over proteins ($^{14}$N/$^{15}$N = 0.2, *Figure 2D*). H3.3 is associated with transcription loci in the chromosome (*Szenker et al., 2011*; *Toyama et al., 2013*) and its frequent replacement by newer H3.3 indicated active transcription and translation activities in the lens, most likely in the capsule and the cortex. By contrast, histone H3.1 and H3.2 in the heterochromatin regions that only renew during cell cycle replication (*Hake and Allis, 2006*) were found to be comprised of a greater proportion of older proteins ($^{14}$N/$^{15}$N = 2.98 and 4.02, respectively), consistent with the notion that most lens fiber cells are postmitotically differentiated and subsequently lose chromatin (*Bassnett and Mataic, 1997*). The overall rate of lens cell growth was estimated to be slow, with 1/5 to 1/4 of H3.2 and H3.1 being synthesized with full $^{15}$N during the 12 week period. However, in these fiber cells that had ceased to replicate, the beaded filament proteins of Phakinin and Filensin (*Blankenship et al., 2001*; *Wenke et al., 2016*) that are implicated in cataract development (*Conley et al., 2000*; *Jakobs et al., 2000*; *Carter et al., 2000*) were still being actively produced ($^{14}$N/$^{15}$N equals 2.02 and 1.02 respectively) (*Figure 2D* and *Supplementary file 1*).

### A wide range of crystallin α, β and γ new protein synthesis

The γ-Crystallins A/E/F/N that are localized to the nuclear lens showed the greatest proportion of their $^{14}$N-proteins ($^{14}$N/$^{15}$N > 20), followed by phosphoglycerate mutase 2 (Pgm2) that catalyzes glycolysis ($^{14}$N/$^{15}$N > 20), and cysteine protease Calpain-3 ($^{14}$N/$^{15}$N = 18.8) responsible for protein degradation (*Figure 2D*). Crystallins are the most abundant proteins in the lens (*Figure 1D*), and they are divided into α, β and γ protein groups based on sequence homology. α- and β- Crystallins (Crya and Cryb) are chaperone proteins for protein refolding under conditions of oxidative stress (*Hejtmancik et al., 2015*; *Horwitz et al., 1999*; *Andley, 2007*), and the densely packed γ-Crystallins (Cryg) contribute to reflection and hardness of the lens. Interestingly, a range of $^{14}$N/$^{15}$N ratios was detected among distinct family members of each class (*Figure 2D* and *Supplementary file 1*): from 1.4 to 3.01 for α-Crystallins, 1.16 to 9.13 for β-Crystallins, and 1.46

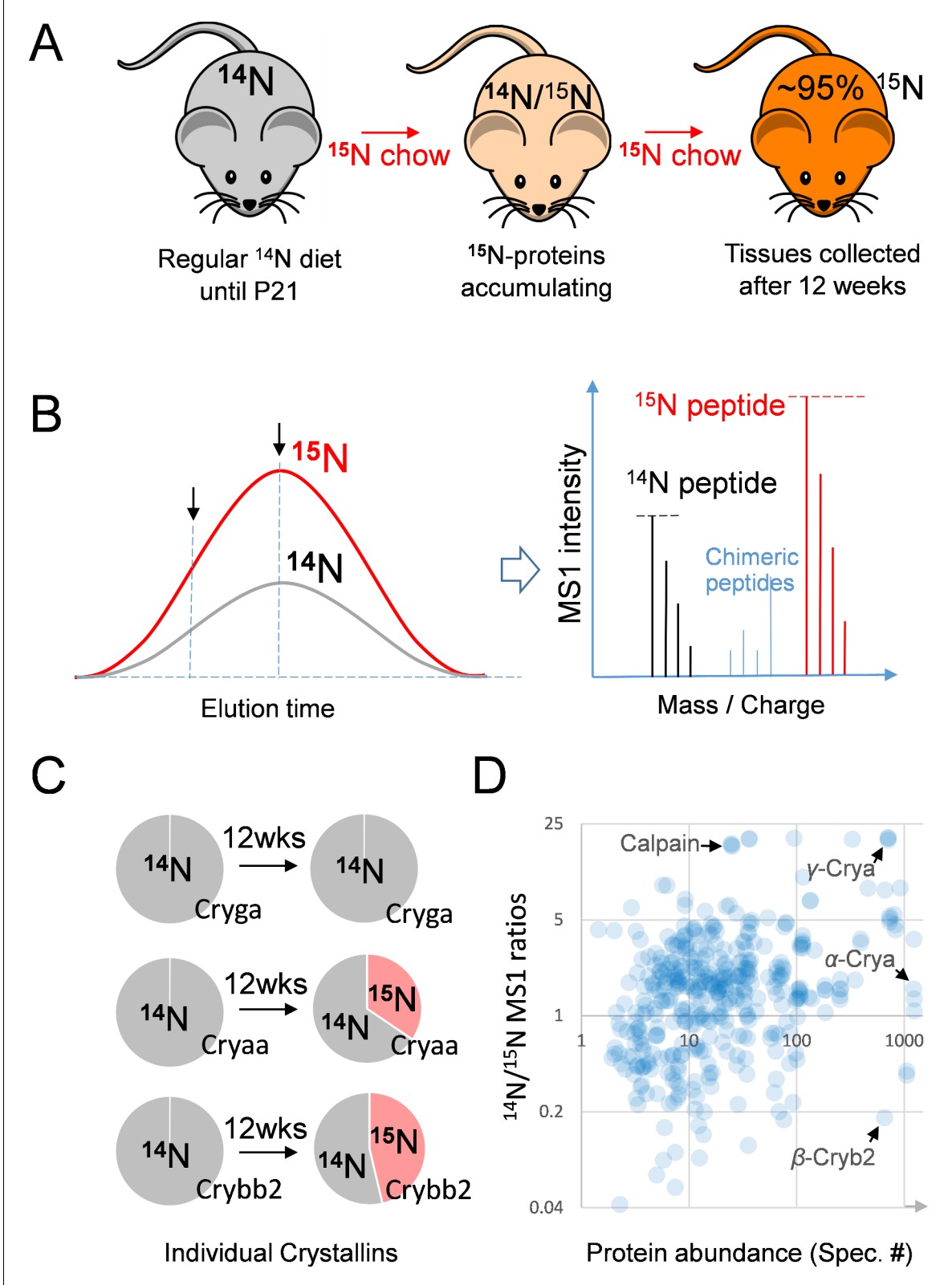

**Figure 1.** The $^{15}$N-labeling workflow for measuring the protein dynamics. (**A**) After weaning, C57BL/6J mice were subjected to an exclusively $^{15}$N chow diet starting at P21 for a total duration of 12 weeks. $^{15}$N was incorporated into newly synthesized proteins. (**B**) In LC-MS/MS, $^{14}$N- and $^{15}$N-peptides of the same sequence co-elute (left panel). Regardless of where MS/MS is triggered (arrows pointing at random positions), the MS1 peptide signal intensities between the $^{14}$N and $^{15}$N channels reflect of their relative abundance (right panel). (**C**) Representative proteins showed different turnover

*Figure 1 continued on next page*

*Figure 1 continued*

rates. D. Among 543 lens proteins (blue dots) that were quantified via their $^{14}$N vs. $^{15}$N ratios (*y*-axis), there was a wide range of total protein abundance as estimated by MS/MS spectral counts (*x*-axis). Note that Calpain protease was long-lived, and the highly abundant α-, β- and γ-Crystallin proteins each had different levels of $^{14}$N and $^{15}$N.

to >20 for γ-Crystallins. These results are in agreement with our previous analysis of Crystallin longevity in aged rats (*Toyama et al., 2013*).

## The gap junctions, the water channels, and the extracellular matrices of the lens

Beside the Crystallins, cataract-linked mutations have been reported in other structural proteins (*Shiels et al., 2010*; *Churchill and Graw, 2011*), including those forming the connexin gap junction channel (*Berthoud and Ngezahayo, 2017*; *Goodenough, 1992*) and aquaporin water channel (*Agre and Kozono, 2003*) (A complete list of all identified proteins in *Supplementary file 1* with selected examples shown in *Figure 2D*). In our dataset, Cx50 was among the longest-lived gap junction proteins with an $^{14}$N/$^{15}$N ratio of 7.19, consistent with Cx50's presence in mature fibers at the nucleus (*White et al., 1992*). This was in contrast with the water channel Aquaporins of Aqp0/Lim1/Mip ($^{14}$N/$^{15}$N = 1.34) (*Bateman et al., 2000*; *Berry et al., 2000*; *Francis et al., 2000*) and Lim2/MP19/Cataract19 ($^{14}$N/$^{15}$N = 3.25) (*Pras et al., 2002*). These gap junctions and Aquaporins form channels known to be critical for the passage of important small metabolites to the lens, and mutations of their genes predispose individuals to cataracts (*Verkman et al., 2014*; *Liu et al., 2011*; *Chepelinsky, 2009*).

Unlike the mature fibers that uniquely form the core of the lens, the outer capsule of the lens resembles other basement membranes such as the glomerular basement membrane of the kidney. Mutations in the major components of type IV Collagen cause Alport syndrome that concurrently affects the kidney, the eye including cataracts and the ear. These type IV Collagen proteins in the lens membrane matrix are produced by the adjacent epithelial cells (*Arita et al., 1993*). Collagen IV-α1, -α2 and -α3 all had balanced $^{14}$N/$^{15}$N ratios of 1.52, 1.34 and 1.34 respectively (*Supplementary file 1*) that had greater proportions of older $^{14}$N-proteins than the majority of lens proteins, however remarkably similar to their counterparts in the kidney at 1.43, 1.43 and 1.16 respectively (not shown). By contrast, another basement membrane protein Perlecan/Hspg2 at the outer and inner surfaces that contributes to anionic charges (*Danysh and Duncan, 2009*) lived longer than Collagen IV ($^{14}$N/$^{15}$N = 4.11) (*Figure 2D*). This apparent contrast of having long-lived Perlecan may partly explain the phenomenon of the lens capsule losing its net anionic charges during aging (*Winkler et al., 2001*): the slow replacement Perlecan may contribute to the gradual loss of its sulfated glycosaminoglycan moieties.

## Contrasting difference between the long preservation of proteins in the nucleus and a varying dynamic turnover of cortex proteins in the lens

Next, we sought to compare protein dynamics in the nucleus and in the cortex of lens. As expected, proteins extracted from the nucleus were mostly shared with their cortex counterparts (*Figure 3A and B*), and a majority of nuclear proteins had little or no protein turnover as determined by their $^{14}$N/$^{15}$N ratios close to or above 100 (*Figure 3B*: upper limit set at 100). Meanwhile, proteins harvested from the cortex tend to have a wide range of new vs. old protein ratios (*Figure 3B*). This was also reflected among Crystallin isoforms (*Figure 3B*: highlighted), with α-Crystallins having the highest contents of $^{15}$N in the cortex (*Figure 3C*). Although all Crystallin isoforms appeared to contain fractions of newly expressed $^{15}$N-proteins in the cortex, their individual abundance at the total protein level vary substantially (*Figure 3C*). For instance, while α- and β-Crystallins had more balanced presence between cortex and nucleus fractions, γ-Crystallin levels in the cortex were very low (*Figure 3C*). When protein abundance of all proteins was compared, the cortex and the nucleus had comparable $^{14}$N levels, in contrast to very low new protein contents in the nucleus (*Figure 3D*, and *Supplementary file 1*).

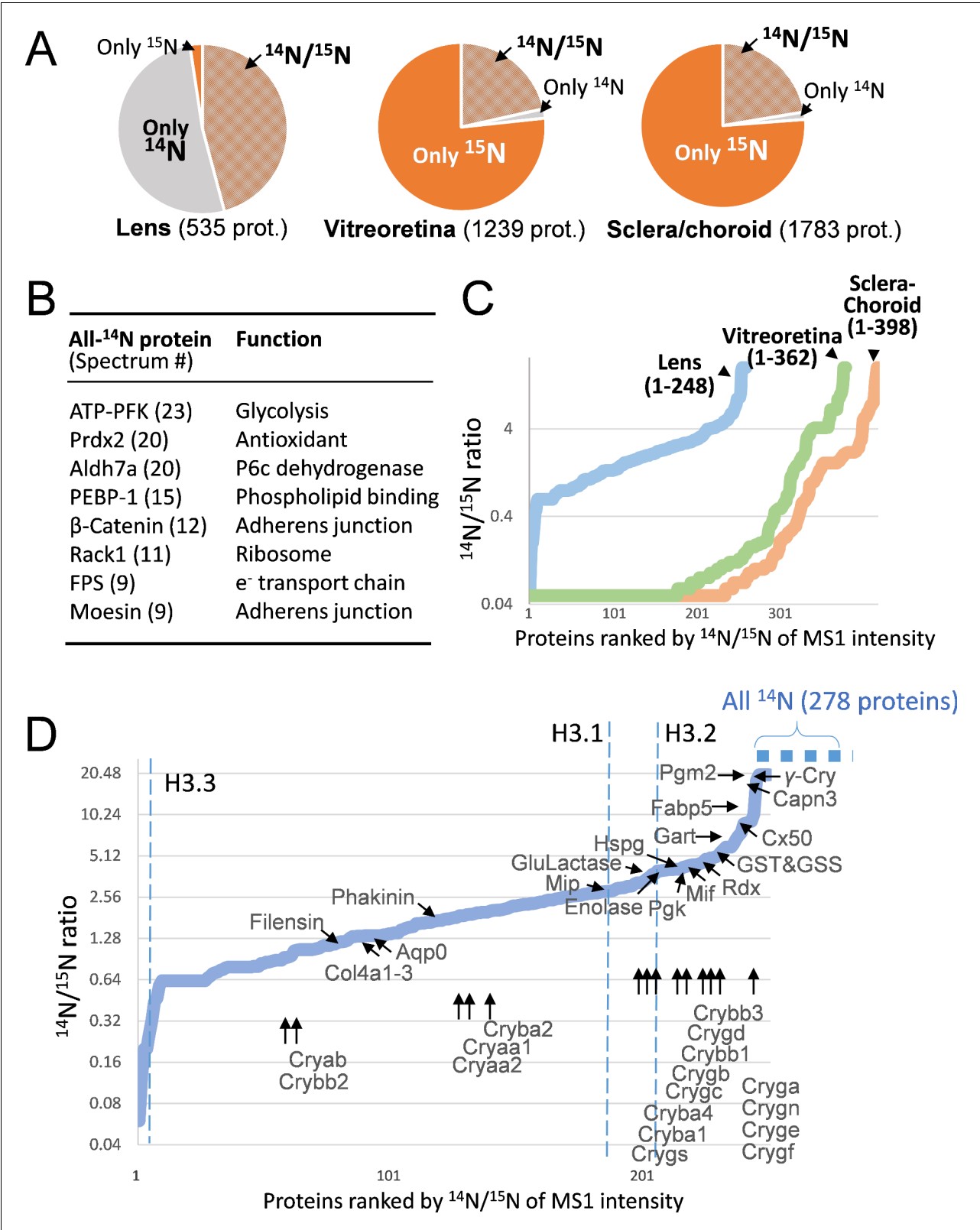

**Figure 2.** The lenticular proteins generally had longer life times than the vitreoretinal and sclera/choroid proteins. (**A**) Inter-tissue comparison of protein longevity showing numbers of proteins based on $^{14}$N-peptides only, $^{15}$N-peptides only, or $^{14}$N/$^{15}$N ratios calculated (hatched pie). While in the lens a large number of protein had a greater proportion of $^{14}$N, other ocular tissues had a faster protein turnover with more proteins completely labeled with $^{15}$N within 12 weeks. (**B**) A list of the most abundant proteins that were only detected by $^{14}$N peptides with the absence of $^{15}$N. (**C**) Among the proteins

*Figure 2 continued on next page*

*Figure 2 continued*

identified with both $^{14}$N and $^{15}$N peptides—248, 362 and 398 proteins in the lens, the vitreoretinal and the sclera/choroid, respectively, the distribution of $^{14}$N-to-$^{15}$N ratios of the lens proteins was different than those of the other tissues. As expected, the lens had the highest proportion of its $^{14}$N-proteins remaining. (D) The distribution of $^{14}$N/$^{15}$N ratios from low to high as in *Figure 2C*. There is a long list of 278 proteins with only their $^{14}$N-proteins detected (dotted blue line to upper right), indicating possibly less $^{14}$N to $^{15}$N conversion of these proteins than those measured with $^{14}$N/$^{15}$N ratios. Proteins that are implicated in cataract including structural proteins, gap junction and water channels, and metabolic enzymes are listed with arrows pointing to their corresponding values. The family of Crystallin proteins are listed below. Benchmark Histones H3.3 and H3.1/3.2 are also listed, representing transcription vs. cell proliferation activities, respectively.

## Very slow turnover of enzymes for Redox and glycolysis

The most unexpected finding of this $^{15}$N-labeling study was that metabolic enzymes in the lens were remarkably long-lived, particularly those catalyzing electron transport chain (ETC) in glycolysis and redox reactions (*Figure 4* and *Supplementary file 1*). For instance, phosphoglycerate mutase 2 (Pgam2) that catalyzes the conversion of 3-phosphoglycerate to 2-phosphoglycerate had an $^{14}$N/$^{15}$N ratio of >20, comparable to those of γ-Crystallin. A number of other enzymes in glycolysis (such as Pgk1,2, 6-Phosphogluconolactone, β-Enolase and γ-Enolase) also had $^{14}$N/$^{15}$N ratios greater than that of histone H3.2 ($^{14}$N/$^{15}$N = 4.02), the benchmark protein that had ceased to renew following fiber cell differentiation. It can be inferred that these enzymes were preserved beyond the point of fiber cell differentiation, which is consistent with the notion that biochemical activity continues in mature fiber cells. Glycolysis that generates electron transport and ATP is an integral process of the overall redox reaction. Given the importance of maintaining a reduced cellular environment, redox enzymes such as glutathione S-transferases (GST) and S-synthases (GSS), and alcohol dehydrogenase class-3 (Adh5) were not only abundantly present in the lens but also long-lived with their $^{14}$N/$^{15}$N ratios between 4.45 and 5.94 after 12 weeks of labeling (*Figure 2D*, *Figure 4* and *Supplementary file 1*). It is important to note that a majority of these enzymes were still preserved in the nuclear lens (*Figure 4* and *Supplementary file 1*) with most of them having predominately $^{14}$N contents. This observation strongly indicates active enzymatic activities at the core of the lens attributable to these extremely long-lasting enzymes.

## Discussion

We performed $^{15}$N-labeling of mice in conjunction with mass spectrometry-based measurement of $^{15}$N- vs. $^{14}$N-protein ratios. These ratios to a great extent reflected the turnover rates of individual proteins in the lens. The results illustrated a range of new protein synthesis activities, as well as an unexpected panel of proteins that were preserved long after terminal differentiation of the fiber cells, particularly in the nuclear lens. In this latter category, besides structural, water channel and chaperone proteins, metabolic enzymes that catalyze glycolysis and redox reactions were long-lived, and therefore may have implications for age-related cataract formation.

Although the mouse model is our convenient choice for the labeling protocol that is associated with a high cost of $^{15}$N-chow, there are notable limitations in addressing the protein basis for cataract. For instance, certain Crystallin isoforms do not express in human (*Figure 3C*). In addition, the choice of rather young animals between 3 and 15 weeks was not ideal for understanding the disease. Instead, the results are more relevant to lens development from post-weaning, through sexual maturity (by 4 weeks) to fully grown adult (by 12–24 weeks), which is equivalent for human age of 20–30 years (information from jax.org) (*Dutta and Sengupta, 2016*). However, the most common presenile cataracts have much later onset. Therefore the protein turnover indices only reveal a narrow spectrum of the changes in protein dynamics in the lens, and the observed preservation of redox enzymes in the nucleus core might not last as long as some Crystallins at old age. In addition to oxidation in cataract lens, many other biochemical changes also occur. Spontaneous conversion of L- to D-amino acids in proteins contributes to racemization in cataract lens (*Hooi and Truscott, 2011*), and protein isomerase activities are thought to have a role counteracting cataract development (*Lyon et al., 2018*; *Lyon et al., 2019*). Related to point that older proteins tend to accumulate post-translational modifications, our mass spec-based approach was not set up to detect all relevant modifications such as non-enzymatic deamidation of Gln and Asn sidechains that occurs more often in old age (*Forsythe et al., 2019*). This omission of modified peptides would have

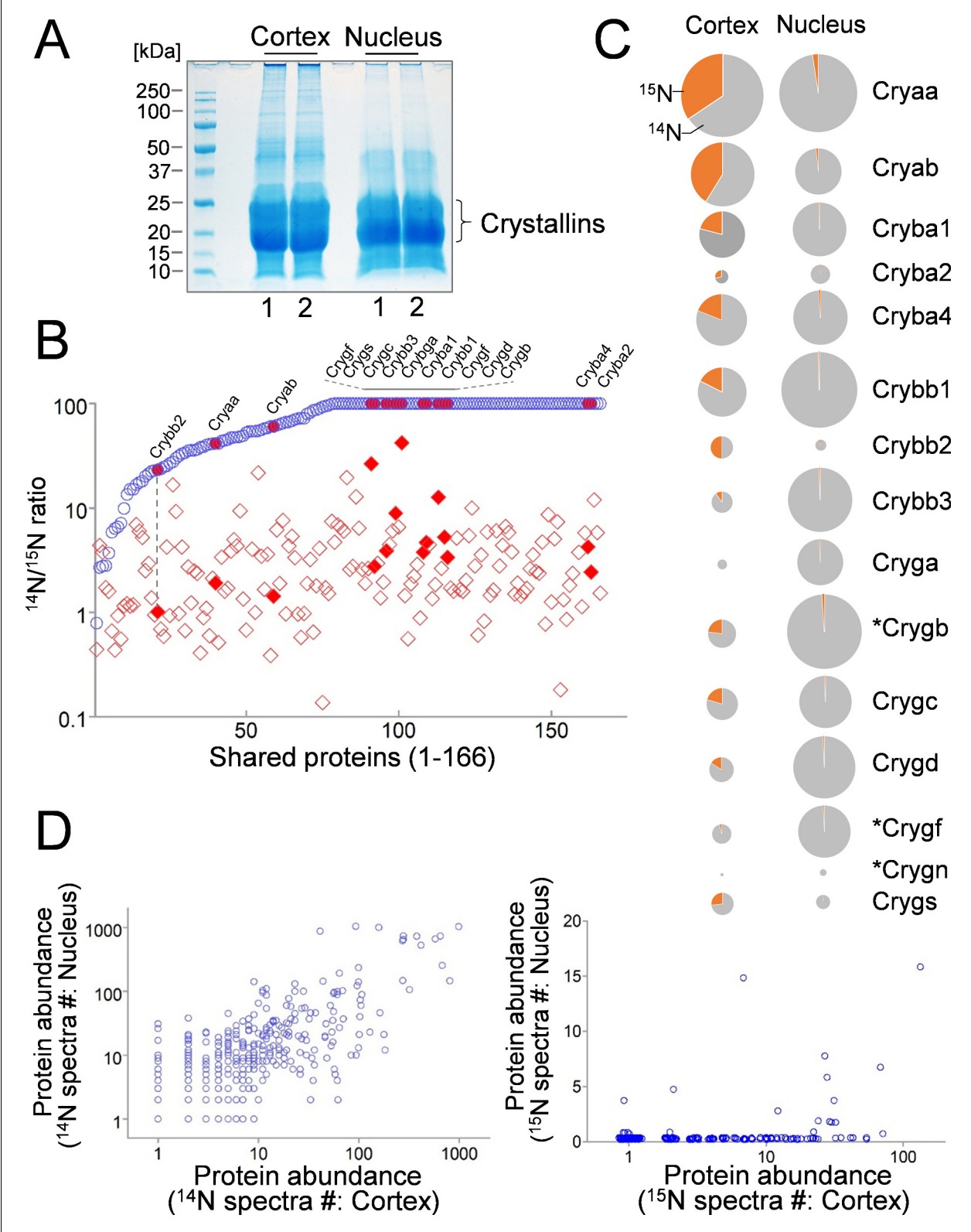

**Figure 3.** Comparison of lens cortex and nucleus proteins by $^{14}N/^{15}N$ ratios. (A) Lens tissues from the the cortex and the nucleus were separately harvested and subsequently resolved by SDS-PAGE. Prominent gel bands of Crystallins were present in both cortex and nucleus fractions, whereas in the higher molecular weight areas of the gel the cortex tissue appeared more intensely stained for its protein contents. (B) A direct comparison of 166 proteins identified in both cortex (red diamond) and nucleus (blue circle) plotted along the x-axis with their $^{14}N/^{15}N$ ratios separately plotted against

*Figure 3 continued on next page*

*Figure 3 continued*

the *y*-axis. In all proteins their $^{14}$N/$^{15}$N ratios were higher in the nucleus (an arbitrary ceiling of the ratios was set at 100 that reflects no protein turnover). There was a wide range of the ratios among individual proteins in the cortex, including all subtypes of Crystalline (filled shapes in red). The ratio values for nuclear Crystallins are also indicated with filled circles at the corresponding *x*-axis positions (example illustrated by the dotted line). (**C**) When individual Crystallin isoforms are compared with respect to their abundance (circle size) in the cortex and the nucleus, there is a general trend of more γ-Crystallins (Cryg) in the nucleus with similar α- and β-Crystallin (Crya and Cryb) contents as compared to the cortex. The $^{14}$N/$^{15}$N ratio indices (pie-chart) among these isoforms in the cortex were also different. Asterisks: Crystallins that are not expressed in human lens. (**D**) Total spectra counts that reflect the relative abundance of individual proteins were plotted for cortex vs. nucleus distributions. Among the old $^{14}$N-proteins there was a relatively balanced distribution between the two fractions (left panel). In contrast, the newer $^{15}$N-proteins were mostly concentrated only in the cortex (right panel).

affected the calculation of $^{14}$N/$^{15}$N ratios, particularly when the types of modifications were more prevalent in the older $^{14}$N-proteins.

Although we only included other non-lens tissues such as the retinal, the sclera and the choroid as controls for having faster turnover dynamics, proteins such as laminin, collagen and fibrillin elastic fibers in these tissues were found to be long-lived (not shown). It should be noted that since we selected a relatively long duration of the labeling process (12 weeks in total), we have passed the most dynamic phase of $^{14}$N-to-$^{15}$N transition in non-lens tissues. It is however anticipated that $^{15}$N pulse-labeling, when the duration is adjusted based on the target tissue, can provide valuable information about protein dynamics, which will be particularly useful in comparing normal and disease tissues for insight on pathogenic transformation or adaptation.

## Materials and methods

### Stable isotope labeling in mouse (SILAM)

The general method of raising $^{15}$N-labeled mice was described previously (*Savas et al., 2012*; *Liu et al., 2018*). In brief, starting at postnatal day 21 after weaning, C57BL/6J mice were fed exclusively with a $^{15}$N-raised spirulina diet (from Cambridge Isotopes and Harlan Laboratories) for 12 consecutive weeks. At this time, the $^{15}$N-proteins in the serum was determined to be greater than 99% by mass spectrometry (*Liu et al., 2018*).

### Harvest of the crystallin lens

Immediately after cervical dislocation of the mice, eye globes were surgically removed and then dissected for collecting the lens. For total protein extraction of the lens, the intact lens was first washed in phosphate-buffered saline, and then submerged into 100 µL of 2x concentrated SDS sample buffer containing 4% SDS, 20% glycerol, 10% 2-mercaptoethanol, 0.004% bromphenol blue and 0.125 M Tris HCl, pH = 6.8. The lens tissue was dissolved following sonication on ice. After the solution turned completely clear, the samples were let to be further dissolved at 4°C for overnight. On the next day, the tissue homogenates from three mice were combined and were resolved by SDS-PAGE. Following staining of the gel with GelCode Blue (Thermo Fisher Scientific), the gel lanes were excised and further divided into ~1 mm$^3$-sized gel cubes. These gel cubes were subsequently subjected to a standard in-gel trypsin/lys-C digestion (Promega), reduction and iodoacetamide alkylation protocol following the manufacturer's instruction.

### Separation of cortex and nucleus tissues

Lenses retrieved from frozen stock formed clear separation between their cortex and nucleus tissues. The cortex in association with the lens capsule were fragile, whereas the nuclear core tissue remained rigid and was picked out using a pair of tissue forceps. The collected nucleus specimens (combined from three eyes) were washed three times in PBS solution before submerged in SDS sample buffer. The lens cortex (also combined from three eyes) was collected without the lens capsule, and then dissolved in SDS sample buffer. Following sonication until the tissue homogenates turned clear, the samples were loaded on an SDSPAGE for protein separation. Proteins were subsequently digested from the gel with Trypsin/LysC as described above.

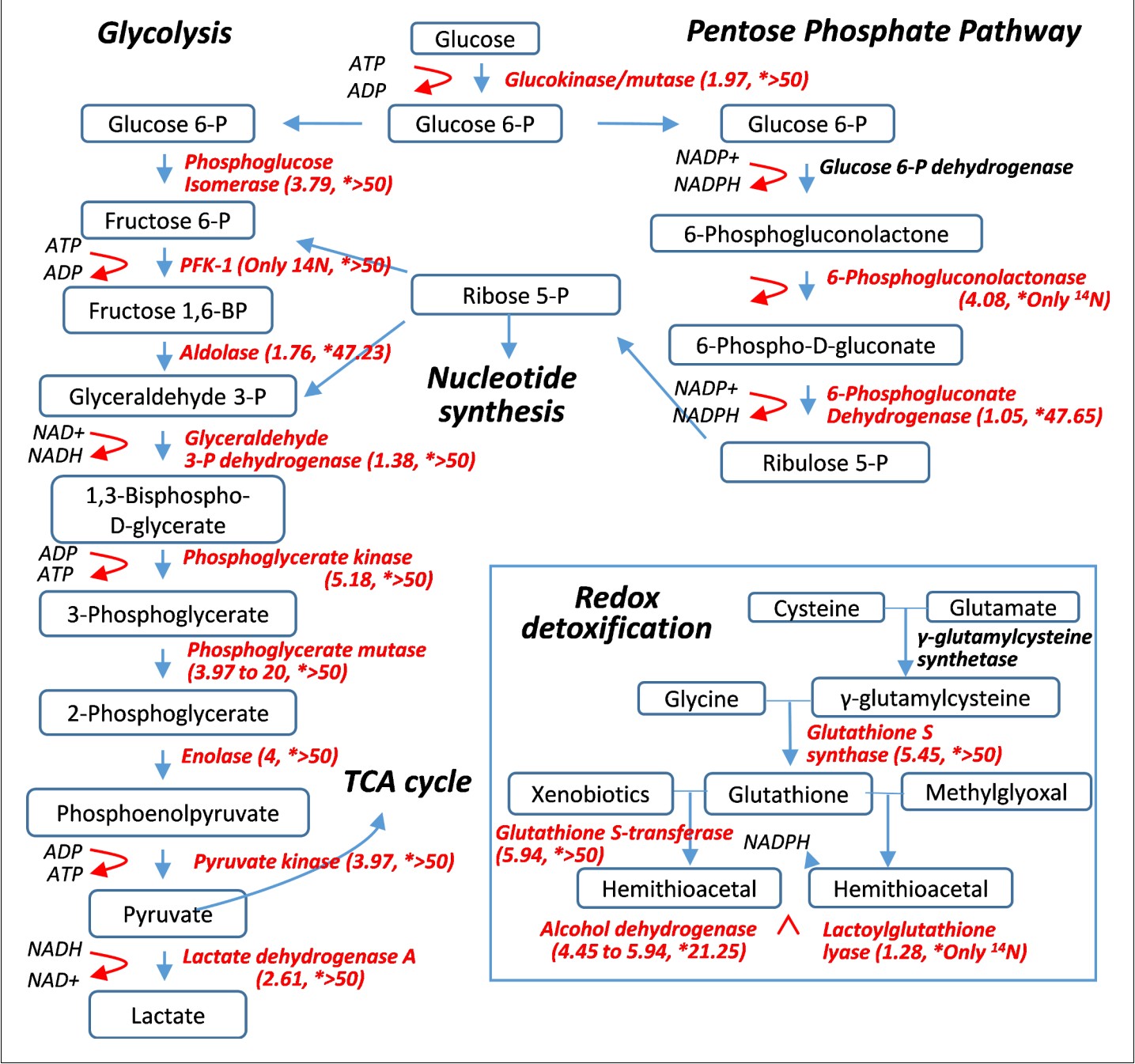

**Figure 4.** Pulse-labeling of the lenticular proteome showed the longevity of metabolic enzymes for redox and glycolysis reactions. Enzymes for energy metabolism and oxidoreduction-detoxification were being measured for their $^{14}$N vs. $^{15}$N abundance in the lens. Marked in red font are proteins that had their $^{14}$N/$^{15}$N ratios successfully measured (with their values in parenthesis; followed by a second set of ratios for their nucleus counterparts marked by asterisks).

## Mass spectrometry

The general procedures for conducting $^{15}$N- vs. $^{14}$N-based proteomics were described previously (*Liu et al., 2018*; *Savas et al., 2017*). In brief, 3 µg of the peptides in Buffer A solution (94.785% H2O, 5% ACN, 0.125% FA) was loaded onto a nanoViper C18 trap column. The peptides were resolved following a 2 hr gradient following an increase Buffer B (99.875% ACN, 0.125% FA) concentration. Peptides were elctrosprayed from the Nanospray Flex Ion Source and analyzed on the Orbitrap Fusion Tribrid mass spectrometer. MS parameters were as follows: ion transfer tube

temp = 300℃, Easy-IC internal mass calibration, default charge state = 2. Detector type set to Orbitrap, with 60K resolution, wide quad isolation, mass range = normal, scan range = 300–1500 m/z. Max injection time = 50 ms, AGC target = 200,000, microscans = 1, S-lens RF level = 60. Without source fragmentation, datatype = positive and centroid, MIPS was on, included charge states = 2–6 (reject unassigned). Dynamic exclusion enabled with n = 1. Precursor selection decision = most intense, top 20, isolation window = 1.6, scan range = auto normal, first mass = 110, collision energy 30%, CID, Detector type = ion trap, max injection time = 75 ms, AGC target – 10,000, inject ions for all available parallelizable time.

## Spectral analysis and protein quantification

Spectral analysis was done using Integrated Proteomics Pipeline (IP2), including running ProLuCID searches against the RefSeq mouse dataset. Basic parameters of 10 ppm precursor mass tolerance and 600 ppm for fragmented ions were used. Searches were filtered with DTAselect containing one peptide per protein, at least one tryptic end and unlimited missed cleavages of a minimum of 6 amino acid, with a false discovery rate (FDR) < 0.001, fixed modification of +57.02146 Da on cysteine residues, and all precursor mass within 10 ppm of expected. To estimate peptide FDRs accurately (set at <1%), target/decoy database was used containing the reversed sequences of all the proteins appended to the target database (*Elias and Gygi, 2007*). Searches were done for combined light and heavy peptides and Census quantified (*Savas et al., 2017*).

To calculate the $^{14}$N/$^{15}$N peptide ion intensity, the ProLuCID results were used to reconstruct MS1 ion chromatograms in the m/z range that included both the heavy and light peptide (*Liu et al., 2018*; *Park et al., 2008*). The intensity ratios were then calculated per peptide using the reconstructed chromatogram. Peptide ratios with correlation values greater than 0.5 were used to remove poor-quality peptide ratio measurements. When more than two peptides were found for the same protein, Census removed outliers based on the Grubbs test (p value < 0.01) by calculating the SDs for the proteins. With QuantCompare, the final peptide ratios were generated. For each protein, its heavy vs. light ratios were represented by the composite of all peptide ratios identified by MS that are assigned to the protein. In cases of extremely low signals in one of the two channels, which will render extremely high or low ratio values mathematically, we arbitrarily set upper and lower limits of the protein ratios at 20 and 0.05. For the cortex vs. nucleus comparison, because of the extreme longevity of nucleus proteins we raised the ceiling to 100 for $^{14}$N/$^{15}$N ratios. The final list of RefSeq protein entries were searched against the UniProtKB database to obtain a non-redundant set of proteins based on their unique gene identifiers (listed in *Supplementary file 1*).

## Acknowledgements

We are grateful to Dr. Amani Fawzi for her suggestions to the study, and Dr. Hongwen Zhou for assisting data analysis. This study was supported by National Institutes of Health (R01AG061787, to JNS, and R01EY025799 and R21AI131087, to JJ).

## Additional information

### Funding

| Funder | Grant reference number | Author |
| --- | --- | --- |
| National Institutes of Health | R01AG061787 | Jeffrey N Savas |
| National Institutes of Health | R21AI131087 | Jing Jin |
| National Institutes of Health | R01EY025799 | Jing Jin |

The funders had no role in study design, data collection and interpretation, or the decision to submit the work for publication.

## Author contributions
Pan Liu, Conceptualization, Data curation, Methodology, Project administration; Seby Louis Edassery, Methodology; Laith Ali, Data curation, Software, Project administration; Benjamin R Thomson, Methodology, Project administration; Jeffrey N Savas, Resources, Supervision, Funding acquisition; Jing Jin, Conceptualization, Resources, Data curation, Supervision, Funding acquisition, Methodology, Project administration

## Author ORCIDs
Pan Liu (iD) http://orcid.org/0000-0002-3066-652X
Benjamin R Thomson (iD) http://orcid.org/0000-0001-6565-5866
Jeffrey N Savas (iD) https://orcid.org/0000-0002-8173-5580
Jing Jin (iD) https://orcid.org/0000-0001-7023-5305

## Ethics
Animal experimentation: All animal procedures were approved by Institutional Animal Care and Use Committee of the Northwestern University (approved protocol number IS00000429 and IS00000862).

## Decision letter and Author response
Decision letter https://doi.org/10.7554/eLife.50170.sa1
Author response https://doi.org/10.7554/eLife.50170.sa2

# Additional files

## Supplementary files
• Supplementary file 1. A complete list of proteins identified by mass spectrometry. Tables S1 to S5 show all proteins identified in whole lens, lens cortex, lens nucleus, sclera choroid and vitreoretinal respectively. The tables also include individual peptide and protein $^{14}N/^{15}N$ ratios, as well as relative total protein amount (calculated as MS spectral count).

• Transparent reporting form

## Data availability
All data generated or analysed during this study are included in the manuscript and supporting files. Source data file (Supplementary File 1) provides a complete list of proteins identified by mass spectrometry. Data has also been deposited at MassIVE under the accession number MSV000084566.

The following dataset was generated:

| Author(s) | Year | Dataset title | Dataset URL | Database and Identifier |
|---|---|---|---|---|
| Liu P, Edassery SL, Ali L, Thomson BR, Savas JN, Jin J | 2019 | Long-lived Metabolic Enzymes in the Crystallin Lens Identified by Pulse-labeling of Mice and Mass Spectrometry | http://doi.org/10.25345/C5H09K | MassIVE MSV0000 84566, 10.25345/C5H0 9K |

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
