## [Decision Letter]

Thank you for submitting your manuscript "Long-lived metabolic enzymes in the crystalline lens identified by pulse-labeling of mice and mass spectrometry" to *eLife*. Your article has been reviewed by two peer reviewers, and the evaluation has been overseen by a Reviewing Editor and Michael Marletta as the Senior Editor. The following individuals involved in review of your submission have agreed to reveal their identity: Ryan Julian (Reviewer #1); Eugene I Shakhnovich (Reviewer #2).

The reviewers have discussed the reviews with one another and the Reviewing Editor has drafted this decision to help you prepare a revised submission.

The reviewers and I were impressed with the work, but each reviewer also had specific and useful comments for improving the manuscript. Specifically, the data interpretation would be substantially clarified if an analysis could be conducted on the separated nucleus and cortex of the lens. Additionally, the paper would be strengthened with a more critical discussion of the data as it relates to longer lived species and to the existing literature on lens and non-lens protein turnover.

I am including the two reviews at the end of this letter. I appreciate that the reviewers' comments cover a range of suggestions for improving the manuscript. We look forward to receiving your revised manuscript.

Reviewer #1:

This manuscript examines protein turnover in the mouse lens by heavy atom labelling followed by mass spectrometric proteomic analysis. The results confirm previous findings; many proteins in the lens are long-lived and not subject to turnover, including enzymes. The results are interesting and certainly worthy of publication, but some of the data is difficult to interpret due to the experimental approach and the work is not adequately placed in the larger context of previous work on long-lived proteins in the lens.

1) The major difficulty here arises from examination of the whole, intact lens, which contains fiber cells in various stages of development. Some of these cells will contain a host of functional organelles and others will not. When combined with the excellent sensitivity of mass spectrometry, the detection of protein turnover becomes difficult to interpret. These proteins may originate primarily from the periphery of the lens where the newest fiber cells are still capable of turnover. Therefore the question of whether fully mature fiber cells allow any protein turnover cannot be answered. Separation of the nucleus and cortex would have allowed analysis of mature fiber cells independently.

2) The timescale should also be explicitly discussed in terms of relevance to the human lens. Although 12 weeks is a long time in the normal sense of protein turnover, it is still not very long compared to 50 years that it might typically take for cataract to occur. Some enzymatic activity could persist for a few weeks, but maybe not over decades.

3) The detection of long-lived proteins outside the lens is interesting and could use more discussion or elaboration in the figures.

4) In relation to cataract, the paper only mentions oxidative markers that correlate with cataract. Recent results have also shown that racemization is also higher in cataract. Hooi and Truscott, 2011. In connection with this, the activity of enzymes (which is mentioned as an unknown) for combating racemization has also been studied as a function of location and therefore longevity within the lens. Lyon, Sabbah and Julian 2018. Cataract is very complicated and connected to many factors. A broader discussion would be useful.

5) It would be valuable to show some of the data for ^14^N/^15^N ratios obtained from different peptides for the same protein to allow readers to evaluate the variation in the data. Perhaps the standard deviation for each value could be given in addition to the average.

Reviewer #2:

(This review was written in collaboration with Eugene Serebryany, a postdoc in the lab who is an expert in crystallins).

This is an interesting and important paper which uses an ingenious isotope labeling approach to determine the lifetimes of proteins in mouse lens in vivo. The findings are not exactly "unexpected" – yes, cytoskeletal proteins and the few enzymes left over in the lens fiber cells don't turn over, because no proteins there turn over – but they seem to be more comprehensive and quantitative than other existing datasets, and nicely presented, too. The Supplementary file 1 that breaks things down by protein is an important resource for future investigators.

There are a couple of important limitations to this study, which the authors should point out explicitly in discussing their results:

First, the lenses were not separated into cortex vs. nucleus. That's a problem, because in neonate and very young mice the outer regions of the lens have not yet completed their differentiation process (not yet enucleated) – see the figure and references in Bassnett, 2002. So, when the authors observe some turnover in crystallins – notably γ-S crystallin, which is preferentially expressed in the cortical region – that could be due to this subpopulation of cells that are still metabolically active, on the background of much more quiescent nuclear cells. Even in adult animals the lens cortex retains some protein turnover, so it would have been really nice to see them separate the lens capsule from cortex from nucleus and mass spec those separately rather than all together.

Second, there are some limitations in how the results from mice translate to humans, and those should be discussed more. Just off the top of my head: the γ-A, E, F, and N crystallins they mention in the paper are present in mice, but humans don't express them.

Finally, the authors should address the following technical yet important problem. It is known that many old proteins, and certainly the lens crystallins, undergo deamidation (when Gln and Asn are gradually, nonenzymatically converted to Glu and Asp). This modification has been found even in quite young lenses, at least in humans. As it happens, deamidation increases molecular weight of the deamidated residue by 1 Da – the same as if an atom of ^15^N were incorporated into the side chain in place of ^14^N. Does the authors' dataset have the resolution to distinguish between deamidated and ^15^N-amide side chains? And have they looked? There is a tiny difference in molecular weight between the two, but many mass spec peak assignment algorithms by default might ignore it. In case they haven't considered deamidation so far, the authors could go back through their data on low-turnover proteins and see whether the apparent ^14^N/^15^N ratios change if all peptides containing Gln or Asn are excluded from the analysis.

---

## [Author Response]

The reviewers and I were impressed with the work, but each reviewer also had specific and useful comments for improving the manuscript. Specifically, the data interpretation would be substantially clarified if an analysis could be conducted on the separated nucleus and cortex of the lens. Additionally, the paper would be strengthened with a more critical discussion of the data as it relates to longer lived species and to the existing literature on lens and non-lens protein turnover.

We have incorporated these suggestions in the revised manuscript with the addition of new experimental results, and included more discussion points. During the revision we focused on generating new data to describe the differences between lens cortex and nucleus. The analysis is summarized in the new Figure 3. In addition, the RAW MS data were deposited to MassIVE database (doi:10.25345/C5H09K).

Reviewer #1:This manuscript examines protein turnover in the mouse lens by heavy atom labelling followed by mass spectrometric proteomic analysis. The results confirm previous findings; many proteins in the lens are long-lived and not subject to turnover, including enzymes. The results are interesting and certainly worthy of publication, but some of the data is difficult to interpret due to the experimental approach and the work is not adequately placed in the larger context of previous work on long-lived proteins in the lens.1) The major difficulty here arises from examination of the whole, intact lens, which contains fiber cells in various stages of development. Some of these cells will contain a host of functional organelles and others will not. When combined with the excellent sensitivity of mass spectrometry, the detection of protein turnover becomes difficult to interpret. These proteins may originate primarily from the periphery of the lens where the newest fiber cells are still capable of turnover. Therefore the question of whether fully mature fiber cells allow any protein turnover cannot be answered. Separation of the nucleus and cortex would have allowed analysis of mature fiber cells independently.

Both reviewers suggested the separation of the cortex and the nucleus. We have conducted new ^15^N-based MS experiments on separately extracted cortex and nucleus samples. We retrieved frozen eye globes from previously labeled mice and subjected them to surgical separation of cortex and nucleus tissues. After thawing, the cortex turned ‘mushy’, and we were able to cleanly separate the nucleus that remained rigid from the cortex. We were also able to collect clean cortex free from cross-contamination from the capsule. It was however difficult to harvest clean capsule without contamination from cortex residues on it. Therefore, we only subjected the nucleus and the cortex for proteomic analysis.

We included the new data in Figure 3, in which we presented both ^14^N/^15^N MS1-based ratio analysis and total spectra count for longevity and overall abundance of each protein, respectively. We have also updated the Supplementary file 1 to include these new results and added new text in the Materials and methods accordingly.

In our revised text:

In Abstract: “Direct comparison of lens cortex versus nucleus revealed little or no ^15^N-protein contents in most nuclear proteins, while there were a broad range of ^14^N/^15^N ratios in cortex proteins. Unexpectedly, like Crystallins, many enzymes with relatively high abundance in nucleus were also exceedingly long-lived.”

In Results:

“Contrasting difference between the long preservation of proteins in the nucleus and a varying dynamic turnover of cortex proteins in the lens.

Next, we sought to compare protein dynamics in the nucleus and in the cortex of lens. As expected, proteins extracted from the nucleus were mostly shared with their cortex counterparts (Figure 3A and 3B), and a majority of nuclear proteins had little or no protein turnover as determined by their ^14^N/^15^N ratios close to or above 100 (Figure 3B: upper limit set at 100). […] When protein abundance of all proteins were compared, the cortex and the nucleus had comparable ^14^N levels, in contrast to very low new protein contents in the nucleus (Figure 3D, and Supplementary file 1).”

And:

“It is important to note that a majority of these enzymes were still preserved in the nucleus of the lens (Figure 4 and Supplementary file 1) with most of them having predominately ^14^N contents. This observation strongly indicates active enzymatic activities at the core of the lens attributable to these extremely long-lasting enzymes.”

2) The timescale should also be explicitly discussed in terms of relevance to the human lens. Although 12 weeks is a long time in the normal sense of protein turnover, it is still not very long compared to 50 years that it might typically take for cataract to occur. Some enzymatic activity could persist for a few weeks, but maybe not over decades.

We added a paragraph to the Discussion section to state the limitations of our mouse study, including the comparison of mouse and human age equivalence.

In our revised text:

In Discussion:

“Although the mouse model is our convenience choice for the labeling protocol that is associated with a high cost of ^15^N-chow, there are notable limitations in addressing the protein basis for cataract. […] Therefore the protein turnover indices only reveal a narrow spectrum of the changes in protein dynamics in the lens, and the observed preservation of redox enzymes in the nucleus core might not last as long as some crystallins at old age.”

3) The detection of long-lived proteins outside the lens is interesting and could use more discussion or elaboration in the figures.

We included a short discussion about long-lived proteins in the sclara, the choroid and the retina. For the reason that a majority of them are extracellular matrix proteins, we didn’t further elaborate them in the figures.

In our revised text:

In Discussion:

“Although we only included other non-lens tissues such as the retina, the sclara and the choroid as controls for having faster turnover dynamics, proteins such as laminin, collagen and fibrillin elastic fibers in these tissues were found to be long-lived (not shown). […] It is however anticipated that ^15^N pulse-labeling, when the duration is adjusted based on the target tissue, can provide valuable information about protein dynamics, which will be particularly useful in comparing normal and disease tissues for insight on pathogenic transformation or adaptation.”

4) In relation to cataract, the paper only mentions oxidative markers that correlate with cataract. Recent results have also shown that racemization is also higher in cataract. Hooi and Truscott 2011. In connection with this, the activity of enzymes (which is mentioned as an unknown) for combating racemization has also been studied as a function of location and therefore longevity within the lens. Lyon, Sabbah and Julian 2018. Cataract is very complicated and connected to many factors. A broader discussion would be useful.

We thank Dr. Julian for pointing out this caveat, and Dr. Shakhnovich for reminding us of another important modification that we will respond to his point below. We have added new discussion on protein racemization and the potential enzymes involved.

In our revised text:

In Discussion:

“In addition to oxidation in cataract lens, many other biochemical changes also occur. Spontaneous conversion of L- to D-amino acids in proteins contributes to racemization in cataract lens (Hooi and Truscott, 2011), and protein isomerase activities are thought to have a role counteracting cataract development (Lyon, Sabbah and Julian, 2018; Lyon et al., 2019).”

5) It would be valuable to show some of the data for ^14^N/^15^N ratios obtained from different peptides for the same protein to allow readers to evaluate the variation in the data. Perhaps the standard deviation for each value could be given in addition to the average.

We revised the Supplementary file 1 to include the ratios of every peptides for each protein to provide the reader a sense of peptide-to-peptide variability. We should also point out that the calculation of the ^14^N/^15^N ratio for a protein is more complicated than simply taking the average of all peptides measured. We used “composite ratio” based on reconstituted MS1 chromatograms that are calculated as under-curve-area of the selected peptides (the quantification software automatically selected peptides with high signal intensity for calculating ratios, as described in Materials and methods). Therefore, the composite ratio reflects a “weighted” average of peptides, taking into account that peptides with higher signal intensity provide more reliable measurements.

Reviewer #2:(This review was written in collaboration with Eugene Serebryany, a postdoc in the lab who is an expert in crystallins).This is an interesting and important paper which uses an ingenious isotope labeling approach to determine the lifetimes of proteins in mouse lens in vivo. The findings are not exactly "unexpected" – yes, cytoskeletal proteins and the few enzymes left over in the lens fiber cells don't turn over, because no proteins there turn over – but they seem to be more comprehensive and quantitative than other existing datasets, and nicely presented, too. The Supplementary file 1 that breaks things down by protein is an important resource for future investigators.

We thank the reviewer for the recognition of our work and we have expanded Supplementary file 1 to include more quantitation information. Two more supplementary tables were also added with new data generated from cortex and nucleus tissues.

There are a couple of important limitations to this study, which the authors should point out explicitly in discussing their results:First, the lenses were not separated into cortex vs. nucleus. That's a problem, because in neonate and very young mice the outer regions of the lens have not yet completed their differentiation process (not yet enucleated) – see the figure and references in Bassnett, 2002. So, when the authors observe some turnover in crystallins – notably γ-S crystallin, which is preferentially expressed in the cortical region – that could be due to this subpopulation of cells that are still metabolically active, on the background of much more quiescent nuclear cells. Even in adult animals the lens cortex retains some protein turnover, so it would have been really nice to see them separate the lens capsule from cortex from nucleus and mass spec those separately rather than all together.

This is an important point, and we included new experimental data separately on the cortex and the nucleus (in new Figure 3 and see our response above to review 1 for further details). In the new analysis, we considered ^14^N/^15^N ratio and protein abundance, and made additional discussion of γ crystallins in our revised manuscript.

In our revised text:

In Discussion:

“For instance, while α- and β-crystallins (Crya and Cryb) had more balanced presence between cortex and nucleus fractions, γ-crystallin (Cryg) levels in the cortex were very low (Figure 3C). When protein abundance of all proteins was compared, the cortex and the nucleus had comparable ^14^N levels, in contrast to very low new protein contents in the nucleus (Figure 3D, and Supplementary file 1).”

Second, there are some limitations in how the results from mice translate to humans, and those should be discussed more. Just off the top of my head: the γ-A, E, F, and N crystallins they mention in the paper are present in mice, but humans don't express them!

We thank you for the comment. We were not aware of the distinctions between mouse and human crystallin genes. We now have these nonhuman crystallin isoforms marked in Figure 3 and included this discussion point in the revised manuscript.

In our revised text:

In Discussion:

“Although the mouse model is our convenient choice for the labeling protocol, which is associated with a high cost of ^15^N-chow, there are a number of limitations in addressing the protein basis for cataract. For instance, certain mouse crystallin isoforms do not express in human (Figure 3C).”

Finally, the authors should address the following technical yet important problem. It is known that many old proteins, and certainly the lens crystallins, undergo deamidation (when Gln and Asn are gradually, nonenzymatically converted to Glu and Asp). This modification has been found even in quite young lenses, at least in humans. As it happens, deamidation increases molecular weight of the deamidated residue by 1 Da – the same as if an atom of ^15^N were incorporated into the side chain in place of ^14^N. Does the authors' dataset have the resolution to distinguish between deamidated and ^15^N-amide side chains? And have they looked? There is a tiny difference in molecular weight between the two, but many mass spec peak assignment algorithms by default might ignore it. In case they haven't considered deamidation so far, the authors could go back through their data on low-turnover proteins and see whether the apparent ^14^N/^15^N ratios change if all peptides containing Gln or Asn are excluded from the analysis.

This is a brilliant comment!

We chose to address it at two different levels in our revised manuscript.

1) We discussed limitation of our approach of not including the search of many post-translational modification types, albeit some are ageing related.

2) We also discussed a technical problem in our calculation of old vs. new ratios without including deamidated peptides that are clear more prevalent in the older ^14^N-proteins.

In our revised text:

In Discussion:

“Related to point that older proteins tend to accumulate post-translational modifications, our mass spec-based approach was not set up to detect all relevant modifications such as non-enzymatic deamidation of Gln and Asn sidechains that occurs more often in old age (Forsythe et al., 2019). This omission of modified peptides would have affected the calculation of ^14^N/^15^N ratios, particularly when the types of modifications were more prevalent in the older ^14^N-proteins.”

In order to precisely assess the impact of protein deamidation on our calculation of ^14^N/^15^N ratios, we performed a separate search for deamidated peptides in the ^14^N channel and have the spectra count values compared to those for unmodified peptides. We summarized the result in Author response image 1, which shows 10.2% peptides being deamidated. This suggest a ~10% underestimation of old ^14^N-protein quantity in our calculation of the ratios.

We also attempted to run searches for ^15^N-peptides for deamidation as a form of modification. However, as the reviewer correctly pointed out, it was challenging. This was because deamidation adds 0.98 atomic mass that is too close to the +0.997 dalton difference between the ^14^N and ^15^N atoms. For this reason we were unable to distinguish deamidation on a ^15^N peptide from its unmodified ^15^N counterpart, given that our MS-MS was performed on a “high-low” setting without the MS2 accuracy for distinguishing 0.997-0.98=0.009 atomic mass.

**Author response image 1. respfig1:** Among all ~600 lens ^14^N-proteins (x-axis), when calculated on the basis of peptide spectra (y-axis), about 10.2% peptides were deamidated (1560 vs. 13703 for deamidated and unmodified peptides).